# Prevalence, determinants, and association of overweight/obesity with non-communicable disease-related biomedical indicators: A cross-sectional study in schoolteachers in Kabul, Afghanistan

**Sharifullah Alemi**[1], **Keiko Nakamura**[1]*, **Ahmad Shekib Arab**[1], **Mohammad Omar Mashal**[1], **Yuri Tashiro**[1], **Kaoruko Seino**[1], **Shafiqullah Hemat**[1,2]

**1** Department of Global Health Entrepreneurship, Division of Public Health, Tokyo Medical and Dental University, Tokyo, Japan, **2** Ministry of Public Health, Kabul, Afghanistan

* nakamura.ith@tmd.ac.jp

**Data Availability Statement:** The relevant data generated and analyzed during this study are within

## Abstract

Overweight/obesity constitutes a major risk factor for non-communicable diseases (NCDs), whose global prevalence is growing rapidly, including in Afghanistan. However, the effects of risk factors on NCDs have rarely been studied in the educator workforce. Therefore, the objective of this study is to determine the prevalence, determinants, and association of overweight/obesity with NCD-related biomedical indicators among schoolteachers in Afghanistan. The sample comprised 600 schoolteachers aged 18 years and above. We conducted questionnaire interviews, anthropometric measurements, and blood biochemistry tests. The main explanatory variable was overweight/obesity (body mass index $\geq$ 25.0 kg/m$^2$). NCD-related biomedical indicators were the outcome variables. Poisson regression models were applied to investigate the association between overweight/obesity and outcome variables. The prevalence of overweight/obesity was 58.2%, which was significantly higher in women, those aged 41–50 years, married participants, and those with 10–20 years of working experience than in their counterparts. After adjusting for sociodemographic variables and lifestyle behaviors, overweight/obesity was significantly associated with hypertension (adjusted prevalence ratio [aPR] = 1.83, 95% confidence interval [CI]: 1.33–2.51); elevated levels of glycosylated hemoglobin (HbA1c) (aPR = 1.35, 95% CI: 1.01–1.79), total cholesterol (aPR = 1.67, 95% CI:1.14–2.44), low-density lipoprotein cholesterol (LDL-C) (aPR = 1.29, 95% CI: 1.10–1.50), and triglycerides (aPR = 1.98, 95% CI: 1.57–2.50), and having three or more comorbidities (aPR = 1.90, 95% CI: 1.47–2.47). Our findings demonstrated a high prevalence of overweight/obesity among schoolteachers. In addition, we found significant associations of overweight/obesity with a higher prevalence of hypertension; elevated serum levels of HbA1c, total cholesterol, LDL-C, and triglycerides; and comorbid conditions in schoolteachers. The findings highlight the need for worksite interventions that promote weight control among schoolteachers with overweight/obesity to reduce the burden of NCDs.

this article and its Supporting Information files (S1 Dataset, S1 Codebook).

**Funding:** This work was funded by the Japan Society for Promotion Science (JSPS) (Grant numbers: 26305022 and 17H02164) granted to K. N. The funder did not have any roles in conceptualization, data collection, analysis, and manuscript writing.

**Competing interests:** The authors have declared that no competing interests exist.

## Introduction

Overweight and obesity are growing public health concerns accounting for at least 4.7 million deaths globally in 2017 [1]. Despite being traditionally considered a concern of high-income countries, overweight and obesity rates among adults have continued to increase in low- and middle-income countries [2]. According to the recent non-communicable disease (NCD) risk factors survey in Afghanistan, the prevalence of hypertension, overweight, obesity, and elevated levels of fasting blood glucose and total cholesterol was 23.5%, 25.8%, 17.0%, 9.2%, and 16.9%, respectively [3]. Previous research analyzing data from a national survey in Afghanistan found that higher age (30 years and over), hypertension, and type 2 diabetes mellitus were among factors positively associated with overweight/obesity [4]. Individuals with overweight/obesity have a greater risk of developing adverse health outcomes, including diabetes mellitus, cardiovascular diseases, cancer, hypertension, and dyslipidemia [5–7]. However, there is a limited understanding of the association between overweight/obesity and NCD-related biomedical indicators. Among the risk factors for NCDs, overweight and obesity are particularly concerning as they potentially reverse many health benefits that would lead to improved life expectancy. The early detection and control of NCD risk factors are regarded as an effective strategy for tackling NCDs. Halting overweight and obesity by promoting healthy lifestyle behaviors, including a balanced diet and regular physical activity, may substantially contribute to achieving the target of reducing NCD-related deaths and disabilities.

Schoolteachers constitute one of the largest and high-risk occupational groups more exposed to the most frequent predictors of overweight and obesity, including poor dietary habits, insufficient physical activity, and spending long working hours on sedentary activities [8–10]. Excess body weight has become more common among employment groups, having negative consequences, including sick leave, more frequent absenteeism and doctor visits, and increased healthcare costs [11–13]. Similarly, participants with overweight/obesity are at risk of functional impairment, early retirement, and reduced health-related quality of life [14–16]. Teachers' teaching quality and productivity may significantly improve when they are healthy, thus having a beneficial impact on students' learning outcomes [17, 18]. Addressing NCDs and their risk factors in schoolteachers contributes to their health outcomes and schoolchildren's educational development and learning outcomes [19, 20]. Thus, investigating health risks among the occupational group of schoolteachers and supporting them to adopt healthy lifestyle behaviors is vital in public health research. Given that schoolteachers spend approximately half of their waking time at school, the school environment and teaching conditions should be reformed to promote and reinforce healthy lifestyle behaviors, such as focusing on food quality, physical activity facilities, and health literacy, which contribute to the prevention of weight gain.

The transition to more urban life and changes in lifestyle behaviors, such as consuming more energy-dense diets and foods high in fat and sugars as well as increases in physical inactivity due to sedentary work/life and modern modes of transportation, have all contributed to increased body weight [21]. Although some research has been carried out on the adverse health outcomes of overweight/obesity among individuals, little is known about the influence of overweight/obesity on increasing the risk of NCDs among schoolteachers. In addition, generating evidence on NCD risk factors among schoolteachers is an important step in designing and developing school-based interventions for the prevention and control of NCDs. This study builds on previous research carried out on the adverse health outcomes of overweight/obesity but differs in that it is the first study to investigate the association between overweight/obesity and a wide range of objectively measured NCD-related biomedical indicators among an important, but rarely-studied occupation group of schoolteachers. Therefore, our study

aimed to determine the prevalence, determinants, and association of overweight/obesity with NCD-related biomedical indicators among schoolteachers in Afghanistan. This study highlights the burden of overweight/obesity as a risk factor for NCD development among adults in Afghanistan. The findings should encourage schoolteachers to modify their lifestyle behaviors to prevent overweight and obesity and reduce the burden of NCDs.

## Methods

### Study design and setting

This was a cross-sectional study conducted in February 2017 that involved 600 schoolteachers from 210 primary, middle, and high public schools across all municipal districts in Kabul city. All permanent male and female schoolteachers were eligible for recruitment. Schoolteachers with short-term contracts and those hired for teachers' training programs were excluded. Based on the formal invitation letter from the Ministry of Education in partnership with the Ministry of Public Health, principals of individual schools were requested to select and introduce one to four schoolteachers who met the eligibility criteria. School principals selected eligible schoolteachers from the list and introduced them to participate in the study. The ratio of male to female schoolteachers in our study is comparable to the sex ratio of schoolteachers in Kabul city. The sample size calculation is described in detail elsewhere [22].

### Data collection and measurements

Data were collected in three phases. First, a face-to-face interview was conducted using a questionnaire that included questions about participants' sociodemographic characteristics, health status, medication history, lifestyle behaviors, and NCD-related knowledge. Second, trained male and female medical staff conducted anthropometric measurements, including height, weight, and blood pressure (BP) measurements. Height (cm) was measured using a stadiometer and weight (kg) using a calibrated weighing scale. Body mass index (BMI) was calculated as follows: weight (kg)/height squared ($m^2$). OMRON monitors (OMRON Healthcare, Kyoto, Japan) were used for BP measurement. The average of two different systolic and diastolic BP readings measured at 3–5-minute intervals was used. After blood pressure measurement, blood samples were drawn by laboratory technicians from all the participants for the blood biochemistry tests, which included measurements of glycosylated hemoglobin (HbA1c), total cholesterol, low-density lipoprotein cholesterol (LDL-C), high-density lipoprotein cholesterol (HDL-C), and triglyceride levels. HbA1c was measured using a fully automated HbA1c analyzer (Clover A1c), and lipid measurements were performed using a Micro-lab 300 semi-automated clinical chemistry analyzer.

### Study variables

The main explanatory variable was overweight/obesity, defined as a BMI $\geq$ 25.0 kg/m². Six NCD-related biomedical indicators were considered dependent variables: BP, HbA1c, total cholesterol, LDL-C, HDL-C, and triglycerides. BP was assessed as a dichotomous, categorical variable (<130/85 mmHg/$\geq$130/85 mmHg). Other binary variables were HbA1c (<5.5%/$\geq$5.5%), total cholesterol (<200 mg/dL/$\geq$200 mg/dL), LDL-C (<100 mg/dL/$\geq$100 mg/dL), HDL-C ($\geq$40 mg/dL/<40 mg/dL), and triglycerides (<150 mg/dL/$\geq$150 mg/dL). Comorbidities included hypertension, elevated HbA1c, high total cholesterol, high LDL-C, low HDL-C, and high triglycerides. The presence of multiple biomedical indicators was categorized into less than three and three or more comorbidities. The cut-off values for normal and elevated blood pressure were in compliance with the categories reported in the 2017 American College

of Cardiology/American Heart Association guidelines for the prevention, detection, evaluation, and management of high blood pressure in adults [23]. The cut-off levels for NCD-related biomedical indicators were set according to clinical practice and guidelines, including the Adult Treatment Panel III (ATP-III) guidelines, systematic reviews, and original studies conducted in countries with similar contexts [24–27]. Sociodemographic variables included sex, age, education attainment, marital status, working experience, and monthly income. Lifestyle-behavior variables included physical exercise/walking, fruit/vegetable consumption, and tobacco use.

## Statistical analysis

Data analyses were performed using Stata software (version 15.1; Stata Corp). The chi-squared test was used to compare the characteristics of the weight-status groups. Considering the high prevalence (>10%) of the binary outcome variables, Poisson regression models were employed [28, 29]. We estimated prevalence ratios (PRs) using Poisson regression models with robust variance to identify correlates of overweight/obesity and investigate the effects of sociodemographic variables and lifestyle behaviors on the relationship between overweight/obesity and NCD-related biomedical indicators. The multivariate models were adjusted for sex, age, education attainment, marital status, working experience, monthly income, physical exercise/walking, consumption of fruits/vegetables, and tobacco use. To investigate the effect modification of sex on the association between the explanatory variable and measured outcomes, a sex-stratified analysis was performed. The sex-stratified models were also adjusted for sociodemographic variables and lifestyle behaviors. The statistical assumptions for the Poisson regression model were checked prior to model fitting. The variance inflation factor (VIF) was computed for the set of independent variables, and only variables with VIF less than 5 were included in the model; multicollinearity between the set of included variables was not observed. The deviance goodness of fit and Pearson goodness of fit tests were also performed using the Stata command "*estat gof*" to assess the overall goodness of fit and adequacy of the Poisson regression model. The test results indicated that the Poisson regression models fit our data well. Statistical significance was set at $P \leq 0.05$.

## Ethical considerations

Ethical approval was obtained from the Tokyo Medical and Dental University Research Ethics Committee and the Afghanistan Ministry of Public Health Institutional Review Board. This research complied with the ethical principles set by the Declaration of Helsinki. All participants were provided with information about the study protocol along with written informed consent forms and the right to not participate or withdraw.

## Results

Table 1 shows the sociodemographic characteristics by weight status of study participants. Two-thirds of the participants (69.3%) were women. Most participants were aged between 41 and 50 years and were married. In the total sample ($n$ = 600), 58.2% were classified as overweight/obese, which was significantly higher among females than in males (64.7% vs. 43.5%). The chi-squared test also revealed that a significantly larger proportion of participants aged 41–50 years old, those currently married, and those with 10–20 years of working experience had overweight/obesity compared to their counterparts.

**Table 1. Overall and weight-status profiles of participants (n = 600).**

| Variables | *n* (%) | Non-overweight/obese (BMI<25.0 kg/m$^2$) | Overweight/obese (BMI≥25.0 kg/m$^2$) | *P* |
|---|---|---|---|---|
| | | (*n* = 251; 41.8%) | (*n* = 349; 58.2%) | |
| Sex | | | | |
| Male | 184 (30.7) | 104 (56.5) | 80 (43.5) | <**0.001** |
| Female | 416 (69.3) | 147 (35.3) | **269 (64.7)** | |
| Age (years) | | | | |
| 18–30 | 142 (23.7) | 90 (63.4) | 52 (36.6) | <**0.001** |
| 31–40 | 148 (24.7) | 59 (39.9) | 89 (60.1) | |
| 41–50 | 191 (31.8) | 56 (29.3) | **135 (70.7)** | |
| ≥51 | 119 (19.8) | 46 (38.7) | 73 (61.3) | |
| Education attainment | | | | |
| 12$^{th}$ grade (high school) graduate | 32 (5.3) | 12 (37.5) | 20 (62.5) | 0.470 |
| 14$^{th}$ grade (2-year college) graduate | 362 (60.3) | 146 (40.3) | 216 (59.7) | |
| College/university graduate or higher | 206 (34.4) | 93 (45.1) | 113 (54.9) | |
| Marital status | | | | |
| Never married | 122 (20.3) | 70 (57.4) | 52 (42.6) | <**0.001** |
| Currently married | 478 (79.7) | 181 (37.9) | **297 (62.1)** | |
| Working experience (years) | | | | |
| <10 | 191 (31.8) | 109 (57.1) | 82 (42.9) | <**0.001** |
| 10–20 | 197 (32.8) | 68 (34.5) | **129 (65.5)** | |
| ≥21 | 212 (35.4) | 74 (34.9) | 138 (65.1) | |
| Monthly income, Afghanis[†] | | | | |
| ≤10,000 | 190 (31.7) | 91 (47.9) | 99 (52.1) | 0.102 |
| 10,001–20,000 | 272 (45.3) | 109 (40.1) | 163 (59.9) | |
| >20,001 | 138 (23.0) | 51 (37.0) | 87 (63.0) | |

Note.

Boldface indicates statistical significance (*p* < 0.05).

[†] Currency exchange: 1 USD = 66.67 Afghanis in January 2017.

## Prevalence of weight status by NCD-related biomedical indicators and lifestyle behaviors

The prevalence of hypertension, elevated HbA1c, and high triglyceride levels were 25.7%, 29.7%, and 42.7%, respectively. Of the participants, 20.2%, 58.7%, and 28.8% had high total cholesterol, high LDL-C, and low HDL-C levels, respectively. Overweight/obesity was significantly more prevalent among participants with hypertension; increased serum levels of HbA1c, total cholesterol, LDL-C, and triglycerides; and low HDL-C levels (Table 2).

## Overweight/obesity and sociodemographic and lifestyle-behaviors

After adjusting for sociodemographic and lifestyle-behavior factors, sex, age, and marital status remained significant correlates of overweight/obesity. Female sex, 31 years of age or over, and being married increased the likelihood of being overweight/obese by 1.48, 1.51, and 1.29 times, respectively (Table 3).

**Table 2. Prevalence of weight status according to non-communicable disease-related biomedical indicators and lifestyle behaviors (n = 600).**

| Variables | n (%) | Non-overweight/obese (BMI<25.0 kg/m2) | Overweight/obese (BMI≥25.0 kg/m²) | P |
|---|---|---|---|---|
| | | (n = 251; 41.8%) | (n = 349; 58.2%) | |
| NCD-related biomedical indicators | | | | |
| Blood pressure (mmHg) | | | | |
| Normal (<130/85) | 446 (74.3) | 213 (47.8) | 233 (52.2) | <0.001 |
| Elevated (≥130/85) | 154 (25.7) | 38 (24.7) | **116 (75.3)** | |
| HbA1c (%) | | | | |
| Normal (<5.5) | 422 (70.3) | 196 (46.4) | 226 (53.6) | <0.001 |
| Elevated (≥5.5) | 178 (29.7) | 55 (30.9) | **123 (69.1)** | |
| Total cholesterol (mg/dL) | | | | |
| Normal (<200) | 479 (79.8) | 219 (45.7) | 260 (54.3) | <0.001 |
| Elevated (≥200) | 121 (20.2) | 32 (26.5) | **89 (73.5)** | |
| LDL-C (mg/dL) | | | | |
| Normal (<100) | 248 (41.3) | 133 (53.6) | 115 (46.4) | <0.001 |
| Elevated (≥100) | 352 (58.7) | 118 (33.5) | **234 (66.5)** | |
| HDL-C (mg/dL) | | | | |
| Normal (≥40) | 427 (71.2) | 193 (45.2) | 234 (54.8) | 0.009 |
| Low (<40) | 173 (28.8) | 58 (33.5) | **115 (66.5)** | |
| Triglyceride level (mg/dL) | | | | |
| Normal (<150) | 344 (57.3) | 183 (53.2) | 161 (46.8) | <0.001 |
| Elevated (≥150) | 256 (42.7) | 68 (26.6) | **188 (73.4)** | |
| Lifestyle behaviors | | | | |
| Physical exercise/walking (per day) | | | | |
| <1 hour | 330 (55.0) | 132 (40.0) | 198 (60.0) | 0.314 |
| ≥1 hour | 270 (45.0) | 119 (44.1) | 151 (55.9) | |
| Consumption of fruits/vegetables (per week) | | | | |
| <4 times | 204 (34.0) | 93 (45.6) | 111 (54.4) | 0.181 |
| ≥4 times | 396 (66.0) | 158 (39.9) | 238 (60.1) | |
| Tobacco use | | | | |
| No | 568 (94.7) | 233 (41.0) | 335 (59.0) | 0.089 |
| Yes | 32 (5.3) | 18 (56.2) | 14 (43.8) | |

Note.

Boldface indicates statistical significance ($p < 0.05$). BMI, body mass index; HbA1c, glycosylated hemoglobin; LDL-C, low-density lipoprotein cholesterol; HDL-C, high-density lipoprotein cholesterol.

## Overweight/obesity and non-communicable disease-related biomedical indicators

Table 4 shows the results of multivariate Poisson regression analyses. After adjusting for socio-demographic and lifestyle-behavior factors, participants with overweight/obesity had a 1.83 times higher likelihood of having hypertension than non-overweight/obese participants (adjusted prevalence ratio [aPR] = 1.83, 95% confidence interval [CI]: 1.33–2.51). Age was a factor that positively influenced the rate of hypertension. Participants with overweight/obesity aged 41 years and over were more likely to have hypertension. On the other hand, participants with overweight/obesity earning a monthly income of more than 20 thousand Afghanis were less likely to have hypertension than their counterparts. Overweight/obesity was significantly associated with high HbA1c levels, with a 1.35 times higher likelihood for participants with

**Table 3. Association of overweight/obesity with sociodemographic and lifestyle-behavior variables (n = 600).**

| Variables | Crude prevalence ratio (95% CI) | P | Adjusted prevalence ratio (95% CI) | P |
|---|---|---|---|---|
| Sex | | | | |
| Male | 1.00 | | 1.00 | |
| Female | **1.49 (1.24–1.78)** | **<0.001** | **1.48 (1.22–1.81)** | **<0.001** |
| Age (years) | | | | |
| 18–30 | 1.00 | | 1.00 | |
| 31–40 | **1.64 (1.27–2.11)** | **<0.001** | **1.51 (1.13–2.02)** | **0.005** |
| 41–50 | **1.93 (1.52–2.44)** | **<0.001** | **1.58 (1.16–2.15)** | **0.003** |
| ≥51 | **1.67 (1.29–2.17)** | **<0.001** | **1.51 (1.08–2.11)** | **0.015** |
| Education attainment | | | | |
| 12th grade (high school) graduate | 1.00 | | 1.00 | |
| 14th grade (2-year college) graduate | 0.95 (0.72–1.26) | 0.747 | 0.89 (0.69–1.16) | 0.402 |
| College/university graduate or higher | 0.88 (0.65–1.18) | 0.387 | 0.84 (0.64–1.11) | 0.230 |
| Marital status | | | | |
| Never married | 1.00 | | 1.00 | |
| Currently married | **1.46 (1.17–1.81)** | **0.001** | **1.29 (1.01–1.64)** | **0.043** |
| Working experience (years) | | | | |
| <10 | 1.00 | | 1.00 | |
| 10–20 | **1.53 (1.26–1.85)** | **<0.001** | 1.14 (0.91–1.43) | 0.252 |
| ≥21 | **1.52 (1.25–1.83)** | **<0.001** | 1.05 (0.82–1.34) | 0.697 |
| Monthly income, Afghanis [†] | | | | |
| ≤10,000 | 1.00 | | 1.00 | |
| 10,001–20,000 | 1.15 (0.97–1.36) | 0.102 | 1.12 (0.95–1.31) | 0.173 |
| >20,001 | **1.21 (1.00–1.46)** | **0.046** | 1.18 (0.98–1.42) | 0.075 |
| Lifestyle behaviors | | | | |
| Physical exercise/walking (per day) | | | | |
| <1 hour | 1.00 | | 1.00 | |
| ≥1 hour | 0.93 (0.81–1.07) | 0.317 | 0.97 (0.85–1.11) | 0.706 |
| Consumption of fruits/vegetables (per week) | | | | |
| <4 times | 1.00 | | 1.00 | |
| ≥4 times | 1.10 (0.95–1.28) | 0.191 | 1.09 (0.94–1.26) | 0.251 |
| Tobacco use | | | | |
| No | 1.00 | | 1.00 | |
| Yes | 0.74 (0.50–1.11) | 0.142 | 0.91 (0.59–1.39) | 0.662 |

Note.

Boldface indicates statistical significance ($p < 0.05$).

[†] Currency exchange: 1 USD = 66.67 Afghanis in January 2017.

The multivariate model is adjusted for sex, age, education attainment, marital status, working experience, monthly income, physical exercise/walking, consumption of fruits/vegetables, and tobacco use.

overweight/obesity than for non-overweight/obese participants (aPR = 1.35, 95% CI: 1.01–1.79). Participants with overweight/obesity aged 31 years or over and those consuming fruits/vegetables more frequently were more likely to have high HbA1c levels than those in other categories. On the other hand, participants with overweight/obesity earning a monthly income of 10–20 thousand Afghanis were less likely to have higher HbA1c levels than their counterparts. Overweight/obesity markedly increased the rate of high total cholesterol (aPR = 1.67, 95% CI: 1.14–2.44). Participants with overweight/obesity aged 41 years or over and those with

**Table 4. Association between overweight/obesity and non-communicable disease-related biomedical indicators by sociodemographic and lifestyle-behavior variables (n = 600).**

| Variables | Adjusted prevalence ratio (95% CI) | | | | | | |
|---|---|---|---|---|---|---|---|
| | Hypertension (≥130/85 mmHg) | Elevated HbA1c (≥5.5%) | Elevated total cholesterol (≥200 mg/dL) | Elevated LDL-C (≥100 mg/dL) | Low HDL-C (<40 mg/dL) | Elevated triglyceride level (≥150 mg/dL) | Comorbidity (≥3 or more) |
| Overweight /obesity (BMI≥25.0 kg/m$^2$) | | | | | | | |
| No | 1.00 | 1.00 | 1.00 | 1.00 | 1.00 | 1.00 | 1.00 |
| Yes | **1.83 (1.33–2.51)** *** | **1.35 (1.01–1.79)** * | **1.67 (1.14–2.44)** ** | **1.29 (1.10–1.50)** ** | 1.23 (0.92–1.63) | **1.98 (1.57–2.50)** *** | **1.90 (1.47–2.47)** *** |
| Sex | | | | | | | |
| Male | 1.00 | 1.00 | 1.00 | 1.00 | 1.00 | 1.00 | 1.00 |
| Female | 0.75 (0.56–1.02) | 1.13 (0.82–1.56) | 1.35 (0.88–2.07) | **1.22 (1.01–1.46)** * | **1.47 (1.04–2.09)** * | **0.76 (0.61–0.96)** * | 1.02 (0.78–1.32) |
| Age (years) | | | | | | | |
| 18–30 | 1.00 | 1.00 | 1.00 | 1.00 | 1.00 | 1.00 | 1.00 |
| 31–40 | 1.88 (0.73–4.87) | **1.68 (1.00–2.82)** * | 1.47 (0.66–3.27) | **1.34 (1.00–1.78)** * | 1.38 (0.79–2.38) | 0.95 (0.65–1.39) | 1.54 (0.92–2.60) |
| 41–50 | **4.07 (1.54–10.75)** ** | **2.53 (1.43–4.48)** ** | **2.44 (1.02–5.87)** * | **1.38 (1.01–1.88)** * | **1.89 (1.04–3.43)** * | 1.16 (0.76–1.75) | **2.26 (1.29–3.95)** * |
| ≥51 | **4.89 (1.78–13.46)** ** | **3.66 (2.00–6.70)** *** | **3.17 (1.27–7.90)** * | **1.40 (1.00–1.95)** * | **2.49 (1.33–4.65)** ** | 1.00 (0.64–1.58) | **2.61 (1.45–4.68)** ** |
| Education attainment | | | | | | | |
| 12$^{th}$ grade (high school) graduate | 1.00 | 1.00 | 1.00 | 1.00 | 1.00 | 1.00 | 1.00 |
| 14$^{th}$ grade (2-year college) graduate | 0.72 (0.48–1.08) | 0.89 (0.58–1.37) | **4.30 (1.09–16.92)** * | 1.23 (0.88–1.72) | 1.92 (0.90–4.11) | 1.21 (0.78–1.88) | 1.22 (0.80–1.87) |
| College/university graduate or higher | 0.75 (0.48–1.18) | 0.82 (0.52–1.31) | 3.84 (0.97–15.19) | 1.21 (0.85–1.70) | 1.65 (0.76–3.60) | 1.19 (0.76–1.87) | 1.15 (0.74–1.79) |
| Marital status | | | | | | | |
| Never married | 1.00 | 1.00 | 1.00 | 1.00 | 1.00 | 1.00 | 1.00 |
| Currently married | 1.06 (0.57–1.98) | 0.80 (0.53–1.21) | 1.17 (0.63–2.16) | 1.11 (0.87–1.41) | 1.33 (0.83–2.14) | 1.19 (0.84–1.68) | 0.92 (0.62–1.37) |
| Working experience (years) | | | | | | | |
| <10 | 1.00 | 1.00 | 1.00 | 1.00 | 1.00 | 1.00 | 1.00 |
| 10–20 | 1.74 (0.93–3.24) | 1.05 (0.69–1.58) | 0.82 (0.42–1.59) | 0.91 (0.72–1.15) | 0.77 (0.50–1.19) | 1.12 (0.80–1.56) | 1.14 (0.77–1.69) |
| ≥21 | 1.52 (0.77–2.99) | 0.97 (0.60–1.57) | 0.94 (0.47–1.89) | 1.11 (0.87–1.42) | 0.83 (0.53–1.31) | 1.18 (0.81–1.71) | 1.12 (0.74–1.71) |
| Monthly income, Afghanis [†] | | | | | | | |
| ≤10,000 | 1.00 | 1.00 | 1.00 | 1.00 | 1.00 | 1.00 | 1.00 |
| 10,001–20,000 | 0.76 (0.58–1.01) | **0.70 (0.53–0.92)** * | 1.04 (0.72–1.50) | 0.91 (0.78–1.05) | 0.86 (0.64–1.14) | 1.03 (0.84–1.26) | 0.90 (0.72–1.13) |
| >20,001 | **0.70 (0.49–0.99)** * | 0.86 (0.63–1.19) | 1.09 (0.70–1.69) | 1.01 (0.85–1.21) | 1.02 (0.72–1.42) | 0.97 (0.75–1.26) | 0.95 (0.72–1.24) |
| Lifestyle behaviors | | | | | | | |
| Physical exercise/ walking (per day) | | | | | | | |
| <1 hour | 1.00 | 1.00 | 1.00 | 1.00 | 1.00 | 1.00 | 1.00 |
| ≥1 hour | 0.84 (0.64–1.10) | 0.84 (0.65–1.07) | 0.97 (0.71–1.34) | 0.92 (0.81–1.05) | 1.07 (0.84–1.37) | 0.94 (0.78–1.13) | 0.89 (0.73–1.09) |

(*Continued*)

**Table 4.** (Continued)

| Variables | Adjusted prevalence ratio (95% CI) | | | | | | |
|---|---|---|---|---|---|---|---|
| | Hypertension (≥130/85 mmHg) | Elevated HbA1c (≥5.5%) | Elevated total cholesterol (≥200 mg/dL) | Elevated LDL-C (≥100 mg/dL) | Low HDL-C (<40 mg/dL) | Elevated triglyceride level (≥150 mg/dL) | Comorbidity (≥3 or more) |
| Consumption of fruits/vegetables (per week) | | | | | | | |
| <4 times | 1.00 | 1.00 | 1.00 | 1.00 | 1.00 | 1.00 | 1.00 |
| ≥4 times | 1.03 (0.79–1.33) | **1.33 (1.01–1.74)** * | 0.94 (0.68–1.31) | 0.95 (0.83–1.09) | 0.92 (0.71–1.20) | 0.95 (0.79–1.14) | 0.91 (0.74–1.12) |
| Tobacco use | | | | | | | |
| No | 1.00 | 1.00 | 1.00 | 1.00 | 1.00 | 1.00 | 1.00 |
| Yes | 1.22 (0.79–1.86) | 1.28 (0.77–2.13) | 1.71 (0.87–3.33) | 1.28 (0.96–1.71) | **1.73 (1.05–2.85)** * | 1.21 (0.89–1.65) | **1.57 (1.10–2.22)** * |

Note.

Boldface indicates statistical significance ($p < 0.05$).

† Currency exchange: 1 USD = 66.67 Afghanis in January 2017. BMI, body mass index; HbA1c, glycosylated hemoglobin; LDL-C, low-density lipoprotein cholesterol; HDL-C, high-density lipoprotein cholesterol; CI, confidence interval.

***$p < 0.001$;

**$p < 0.01$;

*$p < 0.05$

The multivariate models are adjusted for sex, age, education attainment, marital status, working experience, monthly income, physical exercise/walking, consumption of fruits/vegetables, and tobacco use.

a 14[th] grade/2-year college or higher education were more likely to have high total cholesterol. Overweight/obesity was associated with elevated LDL-C levels (aPR = 1.29, 95% CI: 1.10–1.50). The likelihood of elevated LDL-C levels was higher in female participants with overweight/obesity than in male participants with overweight/obesity. Multivariate analysis revealed no association between overweight/obesity and low HDL-C levels. Overweight/obesity was also associated with high triglyceride levels (aPR = 1.98, 95% CI: 1.57–2.50). Female participants with overweight/obesity were less likely to have higher triglyceride levels than male participants with overweight/obesity. Overweight/obesity was associated with a markedly higher likelihood of having three or more comorbidities (aPR = 1.90, 95% CI: 1.47–2.47). Participants with overweight/obesity aged 41 years or older and those using tobacco were more likely to have three or more comorbidities.

## Results of sex-stratified multivariate analysis

The results of sex-stratified multivariate analyses are presented in Table 5. After adjusting for sociodemographic and lifestyle-behavior variables, male participants with overweight/obesity were more likely to have elevated BP; high levels of HbA1c, LDL-C, and triglycerides; low HDL-C levels; and three or more comorbidities than their non-overweight/obese counterparts. On the other hand, female participants with overweight/obesity were more likely to have elevated BP; high levels of total cholesterol and triglycerides; and three or more comorbidities than their non-overweight/obese counterparts. Moreover, multivariate models were applied to check the effect modification in subgroups for other socioeconomic variables, including age, education, and income, and no statistically significant subgroup differences were found.

**Table 5. Association between overweight/obesity and non-communicable disease biomedical indicators stratified by sex according to sociodemographic and lifestyle-behavior variables (n = 600).**

| Variables | Adjusted prevalence ratio (95% CI) | | | | | | |
|---|---|---|---|---|---|---|---|
| | Hypertension ($\geq$130/85 mmHg) | Elevated HbA1c ($\geq$5.5%) | Elevated total cholesterol ($>$200 mg/dL) | Elevated LDL-C ($>$100 mg/dL) | Low HDL-C ($<$40 mg/dL) | Elevated triglyceride level ($\geq$150 mg/dL) | Comorbidity ($\geq$3 or more) |
| **Male stratum** | | | | | | | |
| Overweight /obesity (BMI$\geq$25.0 kg/m$^2$) | | | | | | | |
| No | 1.00 | 1.00 | 1.00 | 1.00 | 1.00 | 1.00 | 1.00 |
| Yes | **1.97 (1.30–2.97)** ** | **1.89 (1.17–3.07)** * | 1.57 (0.79–3.15) | **1.82 (1.35–2.44)** *** | **2.02 (1.18–3.46)** * | **2.44 (1.76–3.39)** *** | **2.86 (1.83–4.46)** *** |
| **Female stratum** | | | | | | | |
| Overweight /obesity (BMI$\geq$25.0 kg/m$^2$) | | | | | | | |
| No | 1.00 | 1.00 | 1.00 | 1.00 | 1.00 | 1.00 | 1.00 |
| Yes | **1.83 (1.12–2.99)** * | 1.14 (0.81–1.61) | **1.60 (1.02–2.51)** * | 1.08 (0.91–1.29) | 0.97 (0.71–1.33) | **1.72 (1.26–2.36)** ** | **1.49 (1.09–2.02)** * |

Note.

Boldface indicates statistical significance ($p < 0.05$). BMI, body mass index; HbA1c, glycosylated hemoglobin; LDL-C, low-density lipoprotein cholesterol; HDL-C, high-density lipoprotein cholesterol; CI, confidence interval.

***$p < 0.001$;

**$p < 0.01$;

*$p < 0.05$

The multivariate models are adjusted for age, education attainment, marital status, working experience, monthly income, physical exercise/walking, consumption of fruits/vegetables, and tobacco use.

## Discussion

Our results demonstrated that overweight/obesity is independently associated with hypertension; higher serum levels of HbA1c, total cholesterol, LDL-C, and triglycerides; and having three or more comorbidities. These findings are particularly concerning, given that over half of the schoolteachers in this study had overweight or obesity. The current prevalence of overweight in Afghan adults is estimated at 25.8%, and that of overweight and obesity combined is 42.8% [3]. The preliminary results of a population-based cross-sectional study in Kandahar province of Afghanistan indicated that the prevalence of overweight and obesity was 32.8% and 31.0%, respectively, and that of central obesity was 63.7%, which was higher in females than males [30]. Overweight/obesity is increasing faster in Afghanistan due to rapid urbanization, changes in dietary patterns, and the tendency of adults to adopt a more sedentary lifestyle.

Several epidemiological studies have documented the association of overweight/obesity with hypertension and its pathological effects on BP [4, 7, 31, 32]. Weight gain, particularly among adults, appears to be a significant risk factor for developing hypertension [33]. The gradual and moderate body weight reduction achieved by regular physical exercise and consumption of low-calorie diets is recommended to normalize BP in hypertensive and normotensive individuals [34]. A modest weight loss, also defined as weight loss of 5%–10% of baseline weight, is regarded as an effective strategy to lose weight and lower BP in individuals with hypertension [34]. A study of adolescents with obesity showed that a weight-loss program comprising diet, behavior change, and exercise resulted in a greater reduction in BP than a

program that only included diet and behavior change [35]. Therefore, it is imperative to educate schoolteachers with overweight/obesity about the effects of weight loss on hypertension and to guide them towards weight reduction. Furthermore, non-overweight/obese schoolteachers must be encouraged to maintain healthy body weight.

Our study also found a significant relationship between overweight/obesity and HbA1c levels. Excess body weight is a leading risk factor for diabetes [5], and weight gain is significantly associated with the risk of diabetes [36]. A previous study in Afghanistan analyzing data from a national survey found a positive association between overweight/obesity and diabetes [4]. Another study conducted in urban areas in Kabul province found that obesity was positively associated with diabetes [37]. Individuals with overweight/obesity have a considerably higher lifetime diabetes risk than healthy individuals [38]. The mechanism is partially explained by the metabolic changes that occur as adipocytes make the body cells less sensitive to insulin, thus altering glucose production in the body. A cohort study found that patients with type 2 diabetes who lost 10% of their body weight after diagnosis were more likely to achieve glycemic control, despite weight regain after four years, than those who had stable weight or weight gain [39]. Lifestyle changes, such as limiting fat intake combined with exercise, tend to be effective in weight loss, thus potentially helping to delay or prevent the onset of diabetes.

A significant association between overweight/obesity and high serum levels of blood lipids was observed in our study. Multiple factors contribute to the pathophysiology of dyslipidemia in obesity that includes increased production of very low-density lipoprotein (VLDL) by the liver, decreased release of triglycerides into the circulation, and failure to trap free fatty acids increased flux of free fatty acids from fat cells to the liver and the formation of small dense low-density lipoprotein particles [40]. A comprehensive lifestyle modification program that includes diet, exercise, and behavior change has been recommended for adults with overweight/obesity to lose weight and lower blood lipid levels, particularly for women, as they have exhibited a higher prevalence of overweight/obesity [41, 42]. Health experts also recommend weight loss to lower BP, hyperglycemia, and elevated levels of blood lipids in individuals with overweight and obesity complicated by hypertension, diabetes mellitus, or dyslipidemia [43].

We observed that female teachers had an increased likelihood of being overweight/obese than their male counterparts. These results are consistent with those of studies conducted in Tanzania, Ghana, and Ethiopia [8, 44, 45]. The sex-stratified analysis revealed that male teachers with overweight/obesity had an increased likelihood of having abnormal levels of all NCD-related biomedical indicators except elevated levels of total cholesterol, whereas female teachers with overweight/obesity had an increased likelihood of having abnormal levels of three of the six indicators than their non-overweight/obese counterparts. Although speculative, a higher prevalence of overweight/obesity in female teachers may predict an increased likelihood of NCD-related biomedical indicators in this group. However, the sex-stratified analysis indicated that male teachers with overweight/obesity are more susceptible to NCD risk. Insights from such modeling can be used to inform current clinical practice and healthcare professional medical advice for male and female patients with overweight/obesity.

Sex differences in overweight/obesity conditions may be explained by physiological and sociocultural mechanisms [46]. The distribution of adipose tissues, their metabolism, and the levels of sex hormones are key physiological mechanisms that vary depending on sex and contribute to variations in body weight and shape [47]. In Afghanistan's context, traditional beliefs and personal attitudes toward body weight may also contribute to the extent of overweight/obesity. Traditionally, fatness has been considered a sign of beauty, superior health, and strength, an attitude that may lead to the consumption of high-energy foods and reduced physical exercise. In addition, the cultural and environmental barriers, such as insufficient single-sex facilities, that prevent women from engaging in physical activity outside the house also

contribute to the tendency of females to have more sedentary lifestyles than males. Further research is needed to assess the attitudes and perceptions of people toward excess body weight and barriers to the reduction of weight or maintaining a healthy weight in Afghanistan. The perception of beauty and body size appears to have been changing in recent years, with global influences and comparisons to models and celebrities supporting the notion that female thinness (i.e., being healthy) tends to be valued in the marriage market [48]. Therefore, traditional and personal beliefs and attitudes should not be overlooked when developing health programs to combat overweight/obesity and its adverse health consequences. These findings indicate further exploration of factors contributing to weight gain in males and females, including social determinants of health, genetics, and environmental factors. Furthermore, the prevalence of overweight/obesity was higher among married adults than unmarried adults. Previous studies have reported similar findings [8, 44]. Married life has been associated with weight-related behaviors. The association between marriage and excess body weight could be due to several factors. According to a cohort study, an increase in body weight among married adults is associated with increased social eating behaviors and consumption of denser foods due to social obligations, which increase the risk of becoming overweight or obese [48].

The results of this study provide a clear picture of the overweight/obesity burden and its role as a major risk factor in a well-educated population. Therefore, effective measures are required to address this public health problem. The Ministry of Public Health is urged to develop and implement well-grounded risk-factor control programs targeting high-risk populations, including schoolteachers. Furthermore, promoting healthy lifestyle behaviors that reduce the prevalence of overweight and obesity, such as consuming a balanced diet and engaging in regular physical exercise, may help to minimize the future burden of NCDs. In the school setting, there is a need to design and implement school-based interventions that incorporate nutrition, physical activity, and sedentary behavior modification. Raising schoolteachers' awareness and knowledge about NCDs and healthy lifestyles through training and counseling will help improve the prevention and control of NCDs among schoolteachers and their students. In addition, creating enabling school environment for schoolteachers and students could provide physical activity opportunities that support weight loss or maintaining a healthy weight.

The strengths of this study included the objective measurement of NCD-related biomedical indicators and BMI, which provide actual and useful data. Furthermore, we measured HbA1c, which is a reliable biomarker for assessing cumulative glycemic history over the past 2–3 months and does not require fasting for several hours before measurement. A relatively homogenous sample of schoolteachers of similar socioeconomic backgrounds minimized variations in education and income levels. This study focused on a topic that has been scarcely addressed among schoolteachers in Afghanistan, and it provided the necessary information for health policy planning and clinical practice. Notwithstanding, the study also had certain limitations. First, the cross-sectional study design did not allow for the determination of a causal relationship. In addition, the self-reported assessment of lifestyle behaviors, including fruit/vegetable intake, physical exercise/walking, and tobacco use, might have been subjected to social-desirability bias and misreporting. Participants are health education volunteers at schools who voluntarily consented to participate in the study among schoolteachers. Targeting all available and eligible teachers at schools during sample selection would thereby minimize selection bias. Therefore, participants in our study may not represent schoolteachers at-large. Finally, caution should be taken when generalizing the study findings; however, we leveraged a large sample of schoolteachers from all districts of Kabul, which includes citizens from various provinces and ethnic groups.

In conclusion, this study demonstrated a relatively high prevalence of overweight/obesity among schoolteachers. We also found statistically significant associations of overweight/obesity with higher prevalence of hypertension; elevated serum levels of HbA1c, total cholesterol, LDL-C, and triglycerides; and comorbid conditions in schoolteachers. These findings highlight that overweight/obesity is a major predictor of NCD burden in schoolteachers in Afghanistan, implying that effective awareness and behavior change interventions are warranted to promote a healthy body weight and lower the risk of NCDs and their complications.

## Supporting information

**S1 Dataset. File contains data used in the analysis.**
(CSV)

**S1 Codebook. This file provides codes for the S1 Dataset indicating variables, labels, and values.**
(DOCX)

## Author Contributions

**Conceptualization:** Sharifullah Alemi, Keiko Nakamura, Shafiqullah Hemat.

**Data curation:** Ahmad Shekib Arab, Mohammad Omar Mashal.

**Formal analysis:** Sharifullah Alemi, Keiko Nakamura, Ahmad Shekib Arab, Mohammad Omar Mashal, Kaoruko Seino, Shafiqullah Hemat.

**Funding acquisition:** Keiko Nakamura.

**Investigation:** Sharifullah Alemi, Ahmad Shekib Arab, Mohammad Omar Mashal, Kaoruko Seino.

**Methodology:** Sharifullah Alemi, Keiko Nakamura.

**Project administration:** Keiko Nakamura, Shafiqullah Hemat.

**Resources:** Keiko Nakamura.

**Software:** Keiko Nakamura.

**Supervision:** Keiko Nakamura, Shafiqullah Hemat.

**Validation:** Keiko Nakamura, Yuri Tashiro, Shafiqullah Hemat.

**Writing – original draft:** Sharifullah Alemi.

**Writing – review & editing:** Keiko Nakamura, Ahmad Shekib Arab, Mohammad Omar Mashal, Yuri Tashiro, Kaoruko Seino, Shafiqullah Hemat.

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
