## [Decision Letter · Decision Letter 0]

8 Sep 2022

PGPH-D-22-01171

Prevalence, determinants, and association of overweight/obesity with noncommunicable disease related biomedical indicators: a case of schoolteachers in Afghanistan

Dear Dr. Nakamura,

Thank you for submitting your manuscript to PLOS Global Public Health. After careful consideration, we feel that it has merit but does not fully meet PLOS Global Public Health’s publication criteria as it currently stands. Therefore, we invite you to submit a revised version of the manuscript that addresses the points raised during the review process.

Please insert comments here and delete this placeholder text when finished. Be sure to:

We look forward to receiving your revised manuscript.

Kind regards,

Zulkarnain Jaafar

Academic Editor

Journal Requirements:

1. Please provide your detailed Financial Disclosure statement. This is published with the article. It must therefore be completed in full sentences and contain the exact wording you wish to be published.

2. We have noticed that you have uploaded Supporting Information files, but you have not included a list of legends. Please add a full list of legends for your Supporting Information files after the references list. 

Additional Editor Comments (if provided):

Reviewers' comments:

Reviewer's Responses to Questions

**Comments to the Author**

1. Does this manuscript meet PLOS Global Public Health’s publication criteria? Is the manuscript technically sound, and do the data support the conclusions? The manuscript must describe methodologically and ethically rigorous research with conclusions that are appropriately drawn based on the data presented.

Reviewer #1: Yes

Reviewer #2: Yes

2. Has the statistical analysis been performed appropriately and rigorously?

Reviewer #1: No

Reviewer #2: Yes

3. Have the authors made all data underlying the findings in their manuscript fully available (please refer to the Data Availability Statement at the start of the manuscript PDF file)?

Reviewer #1: Yes

Reviewer #2: Yes

4. Is the manuscript presented in an intelligible fashion and written in standard English?

Reviewer #1: Yes

Reviewer #2: Yes

5. Review Comments to the Author

Reviewer #1: I congratulate the authors for submitting this manuscript to PLoS Global Public Health. There is a dearth of updated information on the prevalence of NCD-related biomedical indicators in Afghanistan. This manuscript fills a critical knowledge gap.

The manuscript, however, needs substantial revision to be considered for publication. I focused on methodological issues, which are listed under the major comments. I also listed some minor comments for the authors’ consideration.

Major comments:

1. The prevalence of overweight/obesity and NCD-related biomedical indicators was found to be high (p.8, para 4). In the situation where the binary outcome is common, i.e., the prevalence is greater than 10%, the odds ratio (OR) estimated from binary logistic regression overestimates or jeopardizes the true measure of association. In such cases, a prevalence ratio (PR) instead of OR is a suitable option. Authors should consider using Poisson regression models with robust variance, log-binomial, or other advanced methods for calculating the adjusted PR.

2. Both the sub-sections in p.9 (both (this and above) are results from logistic regression analyses. Please consider revising the titles (“Factors associated with overweight/obesity” and “Results of logistic regression analysis”) to make those self-explanatory. Also, please consider explaining it in the “statistics analysis” section so that readers can understand which dependent variable(s) was used for which set of analyses.

3. It is unclear whether the authors checked multi-collinearity, interaction, or effect modification in the data (for other than sex).

Minor comments:

1. Apart from the abstract, line numbers for the manuscript texts were not provided. This would be useful for making specific comments.

2. Both the full title and the short title should consider specifying Kabul, Afghanistan, since the data are from 22 municipal districts of Kabul only.

3. The authors should consider revising the statement (“This was the first study to be conducted on an Afghan population …”) on p.13, para 3 to acknowledge the existing works on similar topics in Afghanistan (viz., Pengpid & Peltzer (2021) doi: 10.1186/s41043-021-00251-0; Saeed (2013) doi: 10.5195/cajgh.2013.69; Saeed et al. (2020) doi: 10.26719/emhj.20.005; Sahrai et al. (2022) doi: 10.1007/s44197-021-00026-0)

4. In Tables 1 and 2, consider replacing “Non-obese” with “Non-overweight/obese.”

Reviewer #2: This cross-sectional study aimed to determine the prevalence, determinants, and association of overweight/obesity with NCD-related biomedical indicators among schoolteachers in Afghanistan. Furthermore, the authors wanted to emphasize the role overweight/obesity in the development of NCDs among school teachers in Afghanistan. This study present evidence-based findings that call for more robust future studies as well as advocate for public health policy to improve adult’s health in Afghanistan. Here, I highlight some of minor and major considerations to be considered prior to publication.

Minor comments

Title: please rewrite the title to read as “Prevalence, determinants, and association of overweight/obesity with noncommunicable diseases-related biomedical indicators: a cross-sectional study in schoolteachers in Afghanistan”.

Introduction

Please add more information about the burden of NCDs and its related risk factors specifically in general population in Afghanistan.

Methods

- Please elaborate more on how the random selection was implemented.

- For the six measured dependents variables, please elaborate more on which bases those specific cut-off values were used to categories those variables into dichotomous variables.

Statistical analysis: please consider writing that ‘to investigate the effect modification of sex on the association between exploratory variable and measured outcomes, sex-stratified analysis was….’

Results

- Please elaborate more on how many were randomly selected and how many consented to participate in this study to reflect the response rate.

- Before addressing the prevalence of overweight/obesity and the deference in that by the measured variables, please provide some descriptive on the collected characteristics of the study population.

- Pleaser replace ‘probability’ with ‘odds’. Probability reelects the risk while odds reflect the likelihood.

Table 2.

I suggest for the sub-categories of the measured biochemical indicators to be written as ‘elevated / normal; whenever applicable. Cut-off values could be provided in brackets for each sub-category.

The column labeled n(%) do not sum up to 600. Please clarify more why there are some missing values in almost all of the measured biomedical indicators. Providing a flow chart presenting number of study participants tested for each of the measured NCD-related biomedical indicator would help.

Overall, the paper reads very well. Minor English editing might be needed. Considering all of the above comments would improve the readability and understandability of the paper when it published.

6. PLOS authors have the option to publish the peer review history of their article (what does this mean?). If published, this will include your full peer review and any attached files.

**Do you want your identity to be public for this peer review?** For information about this choice, including consent withdrawal, please see our Privacy Policy.

Reviewer #1: No

Reviewer #2: No

---

## [Decision Letter · Decision Letter 1]

23 Nov 2022

PGPH-D-22-01171R1

Prevalence, determinants, and association of overweight/obesity with non-communicable disease-related biomedical indicators: a cross-sectional study in schoolteachers in Kabul, Afghanistan

Dear Dr. Nakamura,

Thank you for submitting your manuscript to PLOS Global Public Health. After careful consideration, we feel that it has merit but does not fully meet PLOS Global Public Health’s publication criteria as it currently stands. Therefore, we invite you to submit a revised version of the manuscript that addresses the points raised during the review process.

EDITOR:

Dear Author,

Please attend to all of the reviewers' comments and make the necessary correction or changes.

The decision of this manuscript is justified based on PLOS Global Public Health’s publication criteria and not on its novelty or perceived impact.

We look forward to receiving your revised manuscript.

Kind regards,

Zulkarnain Jaafar

Academic Editor

Journal Requirements:

Additional Editor Comments (if provided):

Reviewers' comments:

Reviewer's Responses to Questions

**Comments to the Author**

1. If the authors have adequately addressed your comments raised in a previous round of review and you feel that this manuscript is now acceptable for publication, you may indicate that here to bypass the “Comments to the Author” section, enter your conflict of interest statement in the “Confidential to Editor” section, and submit your "Accept" recommendation.

Reviewer #1: All comments have been addressed

Reviewer #3: (No Response)

Reviewer #4: (No Response)

2. Does this manuscript meet PLOS Global Public Health’s publication criteria? Is the manuscript technically sound, and do the data support the conclusions? The manuscript must describe methodologically and ethically rigorous research with conclusions that are appropriately drawn based on the data presented.

Reviewer #1: (No Response)

Reviewer #3: No

Reviewer #4: Yes

3. Has the statistical analysis been performed appropriately and rigorously?

Reviewer #1: (No Response)

Reviewer #3: No

Reviewer #4: Yes

4. Have the authors made all data underlying the findings in their manuscript fully available (please refer to the Data Availability Statement at the start of the manuscript PDF file)?

Reviewer #1: (No Response)

Reviewer #3: Yes

Reviewer #4: Yes

5. Is the manuscript presented in an intelligible fashion and written in standard English?

Reviewer #1: (No Response)

Reviewer #3: Yes

Reviewer #4: Yes

6. Review Comments to the Author

Reviewer #1: (No Response)

Reviewer #3: The paper focuses on prevalence of NCDRF among teachers and reports a higher prevalence.

The choice of teachers as a specific group is not well justified. They are definitely of higher SES. What factors in obesity causal pathways are different in teachers. A better justification is required in the introduction.

Introduction is too long and can be shortened to half.

With such high prevalence, is it justified to say the overweight and obesity are "emerging" public health concerns.

Selection of participants for the survey raises serious issues of bias - it seems to be by referral of Principals of the school and not really a random sample.

Why was a poisson regression model used? Will not obesity be a dependent variable in analysis and not explanatory variable?

Not clear why people on treatment for HT/DM/Hyperlipidemia were excluded from analysis. This could have introduced a bias, though the authors justify is that this was done to avoid bias.

The whole issue of female gender as an effect modifier is not well addressed.

Association of obesity with biomedical indicators is well known and does not really add anything.

Table 4 is just too much analysis without any value addition, just because data are available.

Reviewer #4: Reviewer name: Dr Soheir H Ahmed (MD and PhD)

The authors are to be congratulated on an important and useful paper. I have some comments/suggestions that could be used to improve the paper:

1. Authors mentioned:

‘In the total sample (n=600), 58.2% were classified as 180 overweight/obese’.

Question: Is this over all prevalence? Authors need to show their reads separate analysis for mean/prevalence of overweight and obesity for males and females.

2. Authors mentioned:

‘The study team worked with a joint committee of the Afghanistan Ministry of Public Health (MoPH) and the Ministry of Education (MoE) and prepared listwise information on all 210 primary, middle, and high public schools across 22 municipal districts of Kabul. Based on the formal invitation letter from the MoPH and MoE, the principals of individual schools selected and introduced two to three schoolteachers who met the eligibility criteria and consented to participate in the study’.

Question: To my opinion, the methodology of the formation of the study, how the selection was done?

3. Authors mentioned in the discussion (Sex differences in the prevalence of overweight/obesity)

Question: Is there a specific reason for Afghanistan cultural/social why women are more overweight/obese compared to men.

4. Authors did not explain whether all the participants underwent questionaries, anthropometric measurements and blood biochemistry tests?

7. PLOS authors have the option to publish the peer review history of their article (what does this mean?). If published, this will include your full peer review and any attached files.

**Do you want your identity to be public for this peer review?** For information about this choice, including consent withdrawal, please see our Privacy Policy.

Reviewer #1: No

Reviewer #3: **Yes: **Dr. Anand Krishnan

Reviewer #4: No

---

## [Decision Letter · Decision Letter 2]

16 Jan 2023

PGPH-D-22-01171R2

Prevalence, determinants, and association of overweight/obesity with non-communicable disease-related biomedical indicators: a cross-sectional study in schoolteachers in Kabul, Afghanistan

Dear Dr. Nakamura,

Thank you for submitting your manuscript to PLOS Global Public Health. After careful consideration, we feel that it has merit but does not fully meet PLOS Global Public Health’s publication criteria as it currently stands. Therefore, we invite you to submit a revised version of the manuscript that addresses the points raised during the review process.

EDITOR:

Dear Author,

Please attend to all the comments provided by the reviewers and make the necessary corrections.

The decision of this manuscript is justified based on PLOS Global Public Health’s publication criteria and on its novelty or perceived impact.

We look forward to receiving your revised manuscript.

Kind regards,

Zulkarnain Jaafar

Academic Editor

Journal Requirements:

Additional Editor Comments (if provided):

Reviewers' comments:

Reviewer's Responses to Questions

**Comments to the Author**

1. If the authors have adequately addressed your comments raised in a previous round of review and you feel that this manuscript is now acceptable for publication, you may indicate that here to bypass the “Comments to the Author” section, enter your conflict of interest statement in the “Confidential to Editor” section, and submit your "Accept" recommendation.

Reviewer #3: (No Response)

Reviewer #5: All comments have been addressed

2. Does this manuscript meet PLOS Global Public Health’s publication criteria? Is the manuscript technically sound, and do the data support the conclusions? The manuscript must describe methodologically and ethically rigorous research with conclusions that are appropriately drawn based on the data presented.

Reviewer #3: Partly

Reviewer #5: Yes

3. Has the statistical analysis been performed appropriately and rigorously?

Reviewer #3: No

Reviewer #5: Yes

4. Have the authors made all data underlying the findings in their manuscript fully available (please refer to the Data Availability Statement at the start of the manuscript PDF file)?

Reviewer #3: Yes

Reviewer #5: Yes

5. Is the manuscript presented in an intelligible fashion and written in standard English?

Reviewer #3: Yes

Reviewer #5: Yes

6. Review Comments to the Author

Reviewer #3: One needs to encourage publications from Afghanistan. However, the paper still needs to be improved. Issues raised have not been adequately addressed.

It is clear that the sample has a serious selection bias that cannot be corrected now. But they can compare the characteristics of the sample with the teacher population if Afghanistan (if available) For example, 69% were women in the sample, does it hold true for all Afghanistan teachers. Using this estimate to make generalization about all teachers in Afghanistan is inappropriate.

Exclusion of HT/DM on treatment is also not acceptable as it misclassifies individuals. It is not clear why a reanalysis cannot be done with the appropriate classification.

Gender issues does need better exploration. In Discussion, they have just repeated the results and said it needs exploration. If this paper has to add value it has to be in these kinds of issues.

Finding relationship between different metabolic variables is hardly of any significance for Afghanistan or any other country. Table 4 adds little value.

Discussion also has to focus on how to address this problem in schoolteachers as a specific risk group especially school based interventions.

Reviewer #5: The authors have reported a cross sectional study of obesity and associated NCD risk factors among school teachers in Afghanistan.

1. Are there any reasons to believe that obesity and other NCD risk factors are not likely to be related among teachers? Why is studying the teacher population important? Are there occupationally any unique risk factors that do not operate among other occupational groups or the general population? Such justifications will help strengthen the paper.

2. The sampling method has involved the schools selecting and volunteering a few teachers meeting the eligibility criteria from each school. Such volunteer-based sampling, that too done by the school administration is likely to be biased.

7. PLOS authors have the option to publish the peer review history of their article (what does this mean?). If published, this will include your full peer review and any attached files.

**Do you want your identity to be public for this peer review?** For information about this choice, including consent withdrawal, please see our Privacy Policy.

Reviewer #3: **Yes: **Anand Krishnan

Reviewer #5: **Yes: **Vijayaprasad Gopichandran

---

## [Decision Letter · Decision Letter 3]

15 Feb 2023

Prevalence, determinants, and association of overweight/obesity with non-communicable disease-related biomedical indicators: a cross-sectional study in schoolteachers in Kabul, Afghanistan

PGPH-D-22-01171R3

Dear Dr. Nakamura,

We are pleased to inform you that your manuscript 'Prevalence, determinants, and association of overweight/obesity with non-communicable disease-related biomedical indicators: a cross-sectional study in schoolteachers in Kabul, Afghanistan' has been provisionally accepted for publication in PLOS Global Public Health.

Best regards,

Zulkarnain Jaafar

Academic Editor

Reviewer Comments (if any, and for reference):

Reviewer's Responses to Questions

**Comments to the Author**

1. If the authors have adequately addressed your comments raised in a previous round of review and you feel that this manuscript is now acceptable for publication, you may indicate that here to bypass the “Comments to the Author” section, enter your conflict of interest statement in the “Confidential to Editor” section, and submit your "Accept" recommendation.

Reviewer #3: All comments have been addressed

Reviewer #5: All comments have been addressed

2. Does this manuscript meet PLOS Global Public Health’s publication criteria? Is the manuscript technically sound, and do the data support the conclusions? The manuscript must describe methodologically and ethically rigorous research with conclusions that are appropriately drawn based on the data presented.

Reviewer #3: Yes

Reviewer #5: Yes

3. Has the statistical analysis been performed appropriately and rigorously?

Reviewer #3: Yes

Reviewer #5: Yes

4. Have the authors made all data underlying the findings in their manuscript fully available (please refer to the Data Availability Statement at the start of the manuscript PDF file)?

Reviewer #3: Yes

Reviewer #5: Yes

5. Is the manuscript presented in an intelligible fashion and written in standard English?

Reviewer #3: Yes

Reviewer #5: Yes

6. Review Comments to the Author

Reviewer #3: Thanks for making the desired changes

Reviewer #5: Thank you for addressing all my comments appropriately.

7. PLOS authors have the option to publish the peer review history of their article (what does this mean?). If published, this will include your full peer review and any attached files.

**Do you want your identity to be public for this peer review?** For information about this choice, including consent withdrawal, please see our Privacy Policy.

Reviewer #3: **Yes: **Anand Krishnan

Reviewer #5: No
